# Influence of Organ Culture on the Characteristics of the Human Limbal Stem Cell Niche

**DOI:** 10.3390/ijms242316856

**Published:** 2023-11-28

**Authors:** Naresh Polisetti, Gottfried Martin, Eva Ulrich, Mateusz Glegola, Ursula Schlötzer-Schrehardt, Günther Schlunck, Thomas Reinhard

**Affiliations:** 1Eye Center, Medical Center—Faculty of Medicine, University of Freiburg, Killianstrasse 5, 79106 Freiburg, Germany; 2Department of Ophthalmology, University Hospital Erlangen, Friedrich-Alexander-University of Erlangen-Nürnberg, Schwabachanlage 6, 91054 Erlangen, Germany

**Keywords:** limbal stem cells, limbal niche cells, melanocytes, limbal epithelial progenitor cells, human ocular surface, extracellular matrix, cornea, conjunctiva, immunohistochemistry

## Abstract

Organ culture storage techniques for corneoscleral limbal (CSL) tissue have improved the quality of corneas for transplantation and allow for longer storage times. Cultured limbal tissue has been used for stem cell transplantation to treat limbal stem cell deficiency (LSCD) as well as for research purposes to assess homeostasis mechanisms in the limbal stem cell niche. However, the effects of organ culture storage conditions on the quality of limbal niche components are less well described. Therefore, in this study, the morphological and immunohistochemical characteristics of organ-cultured limbal tissue are investigated and compared to fresh limbal tissues by means of light and electron microscopy. Organ-cultured limbal tissues showed signs of deterioration, such as edema, less pronounced basement membranes, and loss of the most superficial layers of the epithelium. In comparison to the fresh limbal epithelium, organ-cultured limbal epithelium showed signs of ongoing proliferative activity (more Ki-67^+^ cells) and exhibited an altered limbal epithelial phenotype with a loss of N-cadherin and desmoglein expression as well as a lack of precise staining patterns for cytokeratin ((CK)14, CK17/19, CK15). The analyzed extracellular matrix composition was mainly intact (collagen IV, fibronectin, laminin chains) except for Tenascin-C, whose expression was increased in organ-cultured limbal tissue. Nonetheless, the expression patterns of cell–matrix adhesion proteins varied in organ-cultured limbal tissue compared to fresh limbal tissue. A decrease in the number of melanocytes (Melan-A^+^ cells) and Langerhans cells (HLA-DR^+^, CD1a^+^, CD18^+^) was observed in the organ-cultured limbal tissue. The organ culture-induced alterations of the limbal epithelial stem cell niche might hamper its use in the treatment of LSCD as well as in research studies. In contrast, reduced numbers of donor-derived Langerhans cells seem associated with better clinical outcomes. However, there is a need to consider the preferential use of fresh CSL for limbal transplants and to look at ways of improving the limbal stem cell properties of stored CSL tissue.

## 1. Introduction

The limbal epithelial stem/progenitor cell (LEPC) niche is located at the corneoscleral limbus and is responsible for the homeostasis of the corneal epithelium, which is an important prerequisite for corneal transparency and visual function [1]. The LEPC niche is characterized by radially oriented epithelial crypts flanked by highly vascularized stromal papillae, called palisades of Vogt [2]. In these special structures, LEPC quiescence, proliferation, and lineage differentiation are maintained in balance through complex interactions with non-epithelial niche cells, extracellular matrix components (ECM), blood vessels, and nerves [3,4,5,6,7]. The limbal tissue has a specialized basement membrane (BM) and ECM protein composition, which contributes to the maintenance of the LEPC phenotype and influences the amount and composition of cytokines and growth factors acting on LEPCs [8,9,10,11]. Similarly, interactions between LEPCs and supporting niche cells, which mainly include limbal melanocytes (LM) and limbal mesenchymal stromal cells (LMSC), are essential for the establishment and maintenance of the niche architecture and for the transmission of regulatory signals to control cell division in the niche [7,12,13]. Furthermore, both LMSC and LM have been shown to have potent anti-inflammatory, immunomodulatory, and anti-angiogenic properties, making them attractive for clinical use as therapeutics [14,15,16,17,18].

A dysfunction or depletion of LEPC, in combination with the destruction of their niche environment, can lead to pathological conditions termed limbal stem cell deficiency (LSCD). In LSCD, the barrier function of the limbus is compromised, leading to a replacement of the corneal epithelium by conjunctival epithelial cells, vascularization, and opacification, eventually leading to vision loss and blindness [19,20]. Various procedures to treat LSCD have been reported, all with the goal of transplanting a new source of corneal epithelium and the removal of altered epithelium and pannus [21,22]. In cases of bilateral LSCD, allogeneic transplantation of donor tissue is necessary either from a related living donor or, more frequently, from a cadaveric donor (from an eye tissue bank), e.g., as keratolimbal allografts (KLAL) or as penetrating limbo keratoplasty (PLK). The overall success rates of PLK are reported to range from 30 to 68% [23,24], whereas KLAL's success rate is limited to 69% [25]. Because of the number of LEPCs present in transplanted limbal tissue and because the surrounding niche microenvironment is the basis for clinical success, current research activities focus on characterizing the limbal niche components required to reproduce the biological niche in vitro [26,27]. Several factors influence the viability and differentiation status of LEPC, such as donor age, death-to-preservation time of the corneas, storage procedure, and duration [28,29,30].

For the preservation and storage of corneal tissue, three procedures are used: cryopreservation, organ culture, and hypothermia. Eye bank organ culture (Tissue-C medium; 37 °C) is commonly used for corneoscleral limbal (CSL) tissue preservation in European countries, and the CSL tissue is usually stored for up to 4 weeks but successful transplants have also been carried out with tissue preserved for up to 7 weeks [31,32]. The major advantages of organ culture include the opportunity to plan surgery in advance, carry out tissue typing (if necessary), and perform microbiological testing before the release of tissue for clinical use. Due to the scarcity of CSL tissues, researchers have been using organ-cultured CSL tissues, not suitable for transplantation or leftover tissue after posterior lamellar keratoplasty, to study limbal stem cell function and the role of limbal niche components in LEPC homeostasis [10,11,17,18,33]. To better understand the limbal stem cell niche and LEPC homeostasis, limbal tissue closest to native limbal tissue is required. The influence of organ culture on CSL tissue has been documented with a focus on central corneal tissue [34,35,36,37,38], but very few studies have addressed limbal tissue [28,29,36]. Roma-Valera and co-workers observed decreased expression of cytokeratin (CK/KRT) 15, vimentin, and ΔNP63α in organ-cultured limbal tissue, but the study was limited to one biological sample [28]. Huag K and co-workers reported that the dual storage procedure (Optisol GS followed by organ culture) retained the LEPC’s proliferative potential, despite lower expression of ABCG2, p63 and connexin-43 as compared to the Optisol storage method alone [36]. Mason and colleagues [39] observed a significant reduction in cell number, TrkA and ΔNp63 expression, and viability in limbal epithelial cells isolated from organ-cultured tissue compared to fresh tissues, yet the study was confined to the assessment of only two markers. However, a detailed examination of limbal niche components of organ-cultured corneas, such as ECM and niche cells, and their similarity to native limbal tissue has not been thoroughly described.

The purpose of this study is to assess the influence of organ culture on the suitability of organ-cultured CSL tissue for allogenic limbal transplantation as well as for research purposes. Therefore, we compare the limbal stem cell niche characteristics of organ-cultured CSL to fresh (non-cultured) CSL using immunohistochemistry and light and electron microscopy.

## 2. Results and Discussion

### 2.1. Architecture of Organ-Cultured Human Corneal Limbus

Hematoxylin and eosin (H&E) staining of fresh CSL tissue sections revealed 5–10 layers of stratified epithelium with darkly stained limbal basal epithelial cells as well as invaginations of the limbal epithelium into the stroma (Figure 1A). Organ-cultured CSL tissues showed signs of deterioration, such as edema and loss of most superficial layers of the epithelium (Figure 1B(i)–(iv)). In some cases, only a single layer of epithelial cells (Figure 1B(ii)) or a total loss of epithelium was observed in small areas. Darkly stained basal limbal epithelial cells were found occasionally (in one out of four samples, Figure 1B(i)), whereas limbal epithelial invaginations into the stroma were absent in organ-cultured limbal tissues (Figure 1B). When compared to fresh limbal tissues, the thickness of organ-cultured limbal epithelium was reduced by more than 50% (Appendix A). The morphological alterations observed in the epithelium after organ-culture storage were similar to those described in previous studies [28,36]. However, we did not observe any notable differences in the limbal stroma of organ-cultured and fresh CSL tissue in terms of collagen matrix and the presence of vessels (dashed circles, Figure 1A,B). The vasculature (dashed circles) within the fresh limbal tissue exhibited erythrocyte occupancy (pink) in a subset of the examined samples (Figure 1A). In contrast, the luminal spaces of vessels in organ-cultured specimens appeared devoid of erythrocytes, presenting a clear lumen (Figure 1B). Periodic acid Schiff (PAS) staining highlighted the BM of the limbal epithelium (arrows) and was less pronounced in the epithelial BM of organ-cultured limbal tissue (Figure 1C).

The transmission electron microscopy (TEM) analysis of the limbus in organ-cultured tissues revealed substantial structural disparities compared to fresh tissues. In fresh tissues, the limbal epithelium appeared as a well-defined 8–10 layered structure, characterized by clusters of small, roundish, densely packed cells situated at the base of the epithelial papillae (Figure 2i), forming the distinctive limbal palisades of Vogt. However, in stark contrast, the organ-cultured limbal tissue exhibited a notable absence of these palisade structures and consisted of thinned limbal epithelium (three to four layers) with basal cuboidal cells along with flattened superficial cells (Figure 2ii), supporting light microscopy studies. The desquamation of superficial cells (Appendix A) and the occurrence of goblet-like cells in superficial layers (identified in three out of six samples; Appendix A) were evident, accompanied by loosely interconnected epithelial cells (Appendix A). The ongoing existence of goblet-like cells in organ-cultured samples appears to denote a potential gradual shift owing to the indistinct limbal architecture. However, comprehensive investigations are imperative to explore and substantiate this observation. Furthermore, frequent accumulation of glycogen (asterisk, Appendix A; Figure 2iv) and lipid droplets within epithelial cells, particularly in the basal and suprabasal layers, was noted. Melanocytes were visible in two out of the six samples analyzed, with a reduced number of melanocytes and their processes, occasionally leaving remnants of melanosomes (arrows) within the basal epithelial cells of organ-cultured tissue (Figure 2iv) compared to fresh tissues (Figure 2iii). Basal adhesion of limbal epithelial progenitor cells (LEPC) mediated by hemidesmosomes (white arrowheads) to the basement membrane (BM, black arrows) in fresh tissue (Figure 2v), whereas the basement membrane displayed signs of attenuation and discontinuity (white arrows), while a loss of hemidesmosomes was identified in organ-cultured tissues (Figure 2vi). These findings collectively underscore marked alterations in cellular structure and composition within organ-cultured corneas, shedding light on the intricate challenges of maintaining limbal niche integrity under such conditions.

### 2.2. Immunohistochemical Characterization of the Organ-Cultured Corneal Limbus

Selected markers related to the epithelial progenitor/differentiation state, ECM and associated molecules, and niche cell-related markers were evaluated by immunohistochemistry using organ-cultured CSL tissues, with representative images provided. A dashed line represents the epithelial BM if it is not demarcated by stained marker proteins. The immunohistochemistry data were correlated with previously published transcriptional profiling data of both fresh and organ-cultured corneas [37], and data sets were retrieved from the Gene Expression Omnibus (GSE214853; https://www.ncbi.nlm.nih.gov/geo/query/acc.cgi?acc=GSE214853, accessed on 3 July 2023). Because the published transcriptional profiling data were limited to the cornea, correlation was restricted to markers expressed on suprabasal limbal cells, across the corneal epithelium, or upregulated in organ culture.

#### 2.2.1. Epithelial Progenitor/Differentiation State

Various studies have reported that LEPCs express CK15, CK14, CK17/19, p63, P(placental)-cadherin, and N(neural)-cadherin, while being negative for CK12, E(epithelial)-cadherin, and desmoglein 1 (DSG1), which are expressed in the suprabasal or corneal epithelium [10,40,41,42]. In our study, the expression of CK12 (red), a cornea-specific keratin, was restricted to the limbal suprabasal area of both the fresh and organ-cultured limbal tissues (arrows indicate negative limbal epithelial basal cells, Figure 3) in line with a previous study [28]. Expression of the epithelial progenitor marker CK14 was reported in the basal layers of both the limbus and cornea [10,41]. In the present study, the expression of CK14 (red) was found in all layers of the limbal epithelium in the organ-cultured limbal tissue (Figure 3), whereas it was limited to basal cells in the fresh limbal tissue (arrows, Figure 3). The RNA sequencing data analysis confirmed an upregulation of KRT14 expression in organ-cultured corneae (Appendix A). Another epithelial progenitor marker, CK17/19 (arrows, red), was also expressed in all layers of the limbal epithelium in organ-cultured CSL tissues similar to CK14 and in contrast to fresh tissue (Figure 3). The RNA sequencing data also confirmed upregulation of KRT17 and KRT19 in organ-cultured corneas (Appendix A). The absence of CK14^−^ and CK19^−^ cells in organ-cultured limbal tissue might be due to a loss of superficial epithelial layers as well as an increase in CK14 and CK19 expression in suprabasal cells of organ-cultured limbal tissue, which warrants further study. p63α (arrows, red), a putative LEPC marker [43,44], was detected in the basal layers of limbal epithelia, but the density of positive cells was significantly lower in organ-cultured limbal tissues (Figure 3). The proliferation marker Ki-67 (arrows, red) was present in the basal cells of the limbus, but the number of Ki-67^+^ cells was higher in organ-cultured tissues compared to fresh CSL tissues (Figure 3). However, RNA sequencing analysis revealed no differences in expression levels between these conditions (MKI67, Appendix A), implying that either Ki-67 expression is regulated at the translational level or that a high abundance of Ki-67 RNA derived from other cell types masks cell type-specific differences in the bulk sequencing data. The high frequency of Ki-67^+^ cells in organ-cultured limbus tissue suggests the ongoing regeneration of the limbal epithelium in line with a previous study [36]. Expression of CK15 (green arrows) was observed in the basal layers of the limbal epithelia in both organ-cultured and fresh CSL tissues but was rather weak in the organ-cultured limbal tissues (Figure 4). Interestingly, CK15 expression was also observed in the basal cells of the peripheral corneal epithelium of organ-cultured CSL tissue (Appendix A), and RNA sequencing data also confirmed upregulation (KRT15, Appendix A). Moreover, CK15^+^ cells in the organ-cultured limbal tissue were bigger in size and loosely structured compared to fresh limbal tissue (Figure 4). LEPCs are characterized by a smaller diameter (around 10 µm), larger nucleus to cytoplasmic ratio, higher cell densities, higher expression of putative stem cell markers, and the highest clonogenic capacity and most label-retaining (side population) cells compared to suprabasal cells [45,46]. The limbal epithelial cells isolated after a short storage time (hypothermic storage) displayed increased population doubling, colony-forming ability, and wound-healing capacity compared to cells stored for longer [29]. However, for further understanding, the culture characteristics of organ-cultured LEPCs must be compared to the ones of LEPCs isolated from fresh limbal tissue. All of these findings indicate that the loss of suprabasal cells resulted in a basal cell population that is actively pursuing a wound healing process and is loosely structured when compared to fresh limbal tissue.

Interactions between LEPCs and surrounding niche cells are required for the formation and maintenance of niche architecture, as well as the transmission of regulatory signals to govern cell division [7,12,13]. Various cadherin family members, which are cell–cell adhesion molecules, have been demonstrated to mediate stem cell–niche interactions in epidermal, neural, mammary, hematopoietic, and limbal stem cells [10,33,42,47]. The superficial layers of limbal epithelial cells in both the organ-cultured and fresh limbal tissues expressed the differentiation marker E-cadherin (red) with rather weak/no expression in the basal cells (arrows, Figure 4). As reported earlier [33], the basal layers of limbal epithelia showed uniform immunoreactivity for P-cadherin (arrows, green) in both fresh and organ-cultured CSL tissues (Figure 4). N-cadherin, another cell–cell adhesion molecule, was shown to maintain hematopoietic stem cell quiescence by angiopoietin-1/Tie-2 signaling [48]. N-cadherin expression was also reported in limbal stem cells, and the downregulation of N-cadherin in LEPC resulted in reduced clonogenic capacity and the formation of thin epithelial sheets with loss of a limbal epithelial phenotype [42,47]. In the present study, N-cadherin expression (red arrows) was observed in the basal cells of fresh limbal tissue but absent in organ-cultured limbal tissues (Figure 4). This suggests that culture conditions trigger the loss of the potent stem cell marker N-cadherin and that basal cells may have undergone partial differentiation. DSG-1, a family of desmosomal cadherins, are involved in the formation of desmosomes that join cells to one another. In fresh limbal tissue, DSG-1 expression (green) was observed in the suprabasal layers of epithelia but not in the basal layers (arrows), whereas its expression was not detected in organ-cultured limbal tissues (Figure 4). Similarly, RNA sequencing data also confirmed significant DSG1 reduction in organ-cultured compared to fresh corneae (Appendix A). The deterioration of superficial limbal epithelial cells explains the lack of staining for desmoglein in organ-cultured CSL tissues. Overall, organ-culture conditions affect the limbal phenotype, inducing a lack of N-cadherin and desmoglein expression, upregulation of CK14 and CK17/19, and a scattered expression of CK15.

#### 2.2.2. ECM and Associated Molecules

ECM molecules have a significant role as they instruct stem/progenitor cell signaling and homeostasis [49]. Limbal tissue has a specialized BM and ECM protein composition comprised of collagen (Col)IV, Tenascin(TN)-C, fibronectin, and laminin (LN) isoforms, which contribute to limbal stem cell niche function [9,10,50,51]. In the present study, the expression of ColIV(green arrows) was observed in the BM of the limbal epithelium as well as in vessels, irrespective of cell culture conditions (Figure 5), similar to a previous study [28]. Fibronectin localization to limbal BM was also similar in both conditions (arrows, Figure 5), and there was no significant difference in RNA expression between fresh and cultured corneae (Appendix A). We reported previously that the LN chains α2, α3, α5, β1, β2, β3, γ1, γ2, and γ3 are strongly expressed in the limbal BM and that the α5-containing isoforms LN-521 and LN-511 are essential to enable efficient expansion of LEPCs and LMs [11,52]. In the present study, the expression of LN-α3 and -α5 (arrows, green) in the limbus of organ-cultured CSL tissues was similar to that of fresh CSL tissue (Figure 5). TN-C, an ECM glycoprotein, is essential for angiogenesis during wound healing [53]; however, conflicting observations were reported on the location of TN-C expression in the limbal region [54,55]. Maseruka and co-workers reported on tenascin C expression in the corneoscleral interface [54], whereas Schlötzer-Schrehardt and co-workers observed its presence underneath the basal limbal epithelium [55]. Moreover, increased expression levels of TN-C deposits underneath the limbal epithelium have been described in aniridia-associated keratopathy as well as during inflammation, fibrosis, and neovascularization [56]. In the current study, TN-C expression was observed in the corneoscleral interface (dotted line separates the limbus and the sclera; Appendix A) and vessels (arrows heads; Appendix A), as well as in the BM of the posterior limbus (arrows, Appendix A), but not in the anterior limbus of fresh CSL tissues (Figure 5). In organ-cultured tissues, TN-C staining was found beneath the limbal epithelium as well as in the limbal stroma (red, Figure 5). A similar TN-C expression pattern was also seen on paraffin sections (Appendix A). The RNA sequencing data also confirmed the upregulation of TNC in organ-cultured cornea compared to fresh corneal tissues (Appendix A). The induced expression of TN-C in organ-cultured limbal tissues is similar to findings in the pathological corneal limbus (e.g., aniridia) and suggests that organ-cultured tissues may have undergone inflammation, fibrosis, and wound healing. Along with ECM components, we also investigated the expression of cell–matrix anchorage proteins β-dystroglycan and integrins α3 and α6. Dystroglycan, a cell–matrix adhesion molecule, is found in various stem cell niches, including the limbal stem cell niche [9,10,57,58]. In the present study, the expression of β-dystroglycan (green arrows) was detected in the basal cell membranes of the limbus as well as in the stroma of both organ-cultured and fresh CSL tissues (Figure 6). Integrins, ECM-binding cell surface receptors, play a major role in various stem cell niches, including the limbal one, by guiding stem cell niche architecture, regulating stem cell proliferation and self-renewal to maintain stem cells in the niche, and, finally, controlling the orientation of dividing stem cells [10,59]. Integrin α3 staining (green, arrows) was present in the basal aspect of the limbal epithelia of fresh limbal tissue, similar to previous findings [10], whereas its expression was observed both in the basal and lateral aspects of limbal epithelia of organ-cultured limbal tissue (Figure 6). The RNA sequencing data also confirmed the upregulation of ITGA3 in organ-cultured tissues compared to fresh corneal tissues (Appendix A). The hemi-desmosomal integrin α6 (red, arrows) occurred in all limbal basal epithelia of fresh limbal tissue with continuous strong basal membrane staining, rather weak in the organ-cultured limbal tissues (Figure 6). The RNA-sequencing data also showed the downregulation of ITGA6 in organ-cultured corneae compared to fresh corneal tissues (Appendix A). Except for TN-C, the composition of extracellular matrix components was intact, but the pattern of cell–matrix adhesion molecule expression was altered in organ-cultured limbal tissues, indicating a loss of epithelial integrity.

#### 2.2.3. Limbal Niche Cells

Aside from cell–ECM interactions, the LEPC maintains close contact with a variety of cell types such as melanocytes (LM), mesenchymal stromal cells (LMSC), vascular cells, immune cells, and nerves across a fenestrated epithelial BM [10,12,13,60]. These non-epithelial limbal niche cells nourish, protect, and regulate the quiescence, self-renewal, and fate decisions of LEPCs. Nerve terminals are closely associated with the limbal stem cell niche and play an important role in the maintenance of the niche environment [61]. Neurofilament (NF)-L antibody was used to detect nerves, nerve terminals, and axons, and its expression (arrows, green) was observed in the stroma of fresh limbal tissues but rarely observed in organ-cultured limbal tissues (Figure 7). RNA sequencing data also confirmed the low expression of NEFL (neurofilament L) in organ-cultured corneae compared to fresh corneal tissues (Appendix A). It has been reported that vascularization of the limbus appears to have an effect on the long-term outcome of limbo-keratoplasty [62]. Grafts with a high degree of vascularization had a tendency for better graft survival in LSCD, but grafts with no vascularization had preferred outcomes in corneal dystrophies [62]. In the present study, we did not observe any significant differences in limbal vasculature between the samples (von Willebrand factor (VWF), Figure 7). It has been reported that transfection of vimentin in epithelial cells leads to the adoption of mesenchymal shapes. The shape transitions are accompanied by a loss of desmosomal contacts and an increase in cell motility [63]. Roma-Valera and co-workers reported a loss of vimentin expression in the limbal epithelium and stroma with increasing tissue preservation times [28]. On the contrary, similar expression patterns of vimentin were observed in the limbal stroma of all the samples analyzed in the current study (Figure 8). Moreover, vimentin^+^ cells were also observed in the epithelial layers of the fresh limbal tissues (arrows, Figure 8), suggesting the presence of either immune cells or melanocytes, as described earlier [18]. Interestingly, we found vimentin expression in limbal basal epithelial cells of organ-cultured samples (arrows, Figure 8), suggesting that LEPCs acquire a more motile phenotype. Expression of CD44, a hyaluronate receptor involved in cell–cell and cell–matrix interactions, was reported in the basal cells of ocular epithelia, stroma, and vasculature [64]. In the current study, CD44 was found on the plasma membranes of basal epithelial cells and stromal cells of the limbus in both conditions (Figure 8). We reported earlier that CD90 has been used as a marker to isolate LMSC from organ-cultured limbal tissues [18,33]. In the present study, CD90^+^ cells (cyan, arrows, Figure 8) were similar in both conditions. Melanocytes are specialized melanin-producing cells of neural crest origin residing within the basal layer of the limbus. Along with a protective function, LMs also support LEPC proliferation and migration and suppress their differentiation to preserve a progenitor state [17,60]. We recently reported the successful isolation of functional limbal melanocytes using CD117 as a selective marker from organ-cultured limbal tissues [18]. Melanocytes (Melan-A^+^ cells) were observed in the basal layers of the limbal epithelium in both conditions (green, arrows, Figure 8); however, the number of melanocytes was significantly lower in organ-cultured limbal tissue sections compared to fresh limbal tissue sections (Appendix A), supporting TEM studies as described earlier. The diminished presence or absence of melanocytes in the organ-cultured limbal tissue suggests unfavorable culture conditions detrimental to the survival of melanocytes. Langerhans cells (LCs), a subset of dendritic cells, play a role in ocular contact hypersensitivity, corneal transplant rejection, and ocular surface immunosurveillance [65]. An absence of HLA-DR^+^ LCs in the limbal tissue was documented after 14 days of cultivation, regardless of storage conditions [35]. Furthermore, the transplantation of corneal buttons with a storage time of 9.8 ± 4.7 days outperformed the transplantation of tissues with a storage time of 1.9 ± 1.4 days [35], suggesting reduced rejection due to diminished donor LC-induced immune activation. In the current investigation, HLA-DR^+^ cells were seen in both the limbal epithelium (red, arrows) and stroma of fresh tissues (arrows), but only in the limbal stroma of organ-cultured tissue (arrows, Figure 8), similar to previous observations [35]. Similarly, CD18^+^ cells (arrows, cyan), common leukocyte markers, were observed in the limbal stroma as well as in the limbal epithelium of fresh tissues, whereas CD18^+^ cells were only observed in the limbal stroma of organ-cultured tissues (Appendix A). CD1a^+^ Langerhans cells (green, arrow) were observed in the limbal epithelium of fresh limbal tissues but were not found in organ-cultured tissues (Appendix A).

In summary, organ culture altered the limbal epithelial phenotype and limbal cell–matrix adhesion patterns and led to a loss of other limbal niche cells such as melanocytes and Langerhans cells. These alterations may affect the success of cultivated limbal stem cell transplantation and cell-culture-based studies on limbal niche function. In contrast, the use of organ-cultured corneae harboring reduced numbers of donor-derived Langerhans cells may be associated with better clinical outcomes. Considering the potential use of eye bank donor tissue in various limbal epithelial transplant procedures, the effect of commonly used eye bank storage systems on the preservation of stemness deserves further investigation.

## 3. Materials and Methods

### 3.1. Human Tissues

Human donor corneoscleral tissues (*n* = 9 (*n* = 6 for immunohistochemistry (3-paraffin, 3-frozen; *n* = 3 for electron microscopy); mean age 72.4 ± 12.4 years; <16 h after death) not suitable for transplantation and organ-cultured corneoscleral tissue (mean age 70.6 ± 10.3 y, post-mortem duration 26.8 ± 7.8 h; culture duration 31.2 ± 5.5 d; *n* = 14 (*n* = 8 for immunohistochemistry (4-paraffin, 4-frozen); *n* = 6 for electron microscopy) after retrieval of corneal endothelial transplants, with appropriate research consent provided by the Lions Cornea Bank Baden-Württemberg, were used as described previously [33]. The informed consent of the donor or their next of kin to corneal tissue donation was acquired. Experiments using human tissue samples were approved by the Institutional Review Board of the Medical Faculty of the University of Freiburg (25/20) and followed the principles of the Helsinki Declaration.

### 3.2. Tissue Preparation and Histology

Corneoscleral tissue was excised from the eyeball using a scalpel and scissors and processed for histology. Tissue was either fixed in 4% paraformaldehyde and embedded in paraffin for 30 min or embedded and frozen in an optimal cutting temperature (OCT) medium. Sections with a thickness of five micrometers were cut and stained as described previously [33]. Briefly, sections were stained with hematoxylin (Haematoxylin Gill III, Surgipath, Leica, Germany) for 2 min and 1% eosin Y (Surgipath, Leica, Germany) (H&E) for 1 min to observe the gross tissue architecture. To visualize the proteoglycan content, periodic acid Schiff (PAS) staining was performed. PAS staining was performed using 1% periodic acid (Honeywell Fluka, Charlotte, NC, USA) for 10 min and Schiff reagent (Roth, Karlsruhe, Germany) for 90 s. Samples were examined using either a Hamamatsu NanoZoomer S60 (Hamamatsu Photonics, Herrsching, Germany) or a bright field fluorescence microscope (Axio Imager.A1, Zeiss, Oberkochen, Germany), and images were processed using ProgRes CapturePro Software V1.1.10.6 (JENOPTIK, Jena, Germany).

### 3.3. Immunostaining of Frozen Sections

Immunostaining of frozen sections was performed as previously described [18]. Briefly, CSL tissue in OCT medium was cut into 6–8 µm sections, fixed in 4% paraformaldehyde (PFA) for 20 min, followed by permeabilization in 0.3% Triton X-100 in PBS for 10 min. The sections were blocked with 10% normal goat serum (NGS) or 10% normal donkey serum (NDS) and then incubated with primary antibodies (Appendix A) diluted in 1% NGS or NDS in PBS overnight at 4 °C or 2 h at room temperature. Alexa-fluor-488, -568, or -647-conjugated anti-mouse or -rabbit immunoglobulins (Life Technologies, Carlsbad, CA, USA) were used for detection and nuclear staining was performed with 4′,6-diamidino-2-phenylindole (DAPI) (Vectashield antifade mounting medium with DAPI; Vector, Burlingame, CA, USA). Immunolabeled cryosections were examined with a laser scanning confocal microscope (TCS SP-8, Leica, Wetzlar, Germany). For negative controls, the primary antibodies were replaced by equimolar concentrations of an irrelevant isotypic primary antibody of the same species.

### 3.4. Immunostaining of Paraffin Sections

Immunohistochemistry was performed on paraffin sections, as previously described [66]. The list of antibodies is provided in Appendix A.

### 3.5. Transmission Electron Microscopy

Both fresh (*n* = 3) and organ-cultured CSL tissue (*n* = 6) specimens were processed for transmission electron microscopy (TEM), as described previously [55]. Briefly, the samples were fixed in 2.5% glutaraldehyde in 0.1 M phosphate buffer, dehydrated, and embedded in epoxy resin according to standard protocols. Ultrathin sections were stained with uranyl acetate–lead citrate and examined with an electron microscope (EM 906E; Carl Zeiss Microscopy, Oberkochen, Germany).

## Figures and Tables

**Figure 1 ijms-24-16856-f001:**
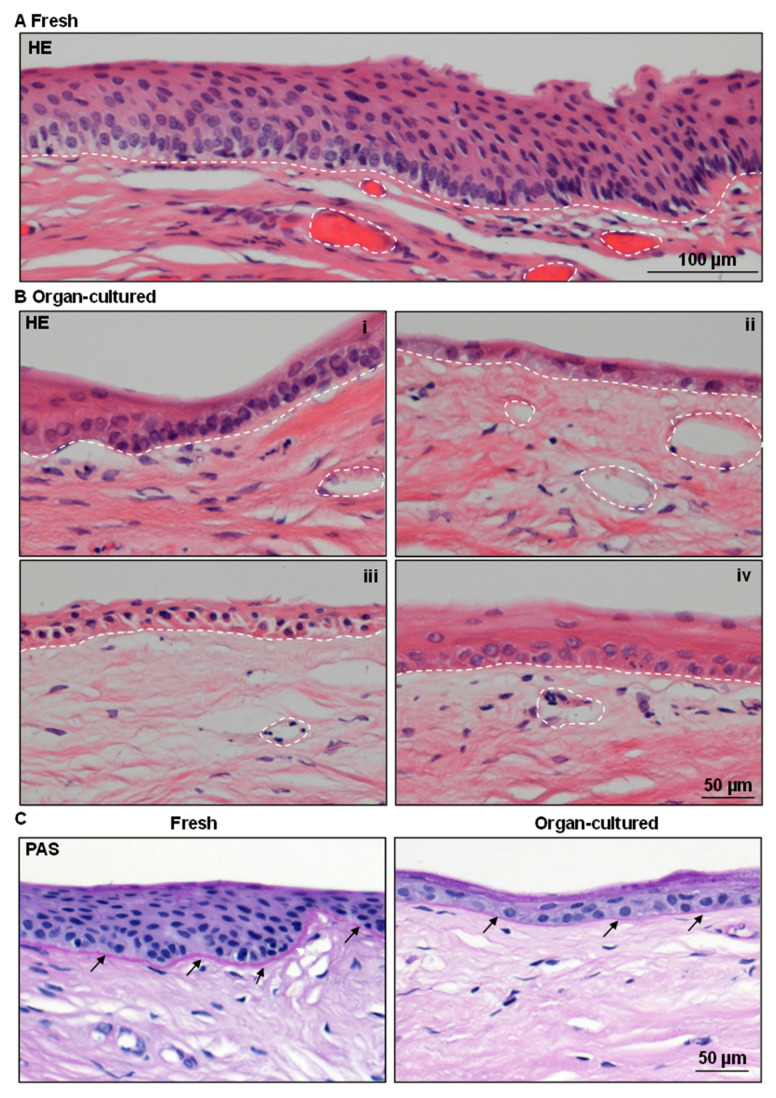
Architecture of organ-cultured human corneal limbus: (**A**) Hematoxylin and eosin (H&E) staining of fresh limbal tissue sections showing multilayered stratified epithelium with darkly stained limbal basal epithelial cells as well as invaginations of the limbal epithelium into the stroma. (**B**) Organ-cultured limbal tissues showing signs of deterioration, such as edema and loss of most superficial layers of the epithelium ((**i**)–(**iv**)). (**C**) Periodic acid Schiff (PAS) staining highlights the basement membrane (BM) of the limbal epithelium (arrows) in fresh limbal tissues and is less pronounced in the organ-cultured limbal tissue. The dashed line represents the BM. Dashed circles represent the vasculature.

**Figure 2 ijms-24-16856-f002:**
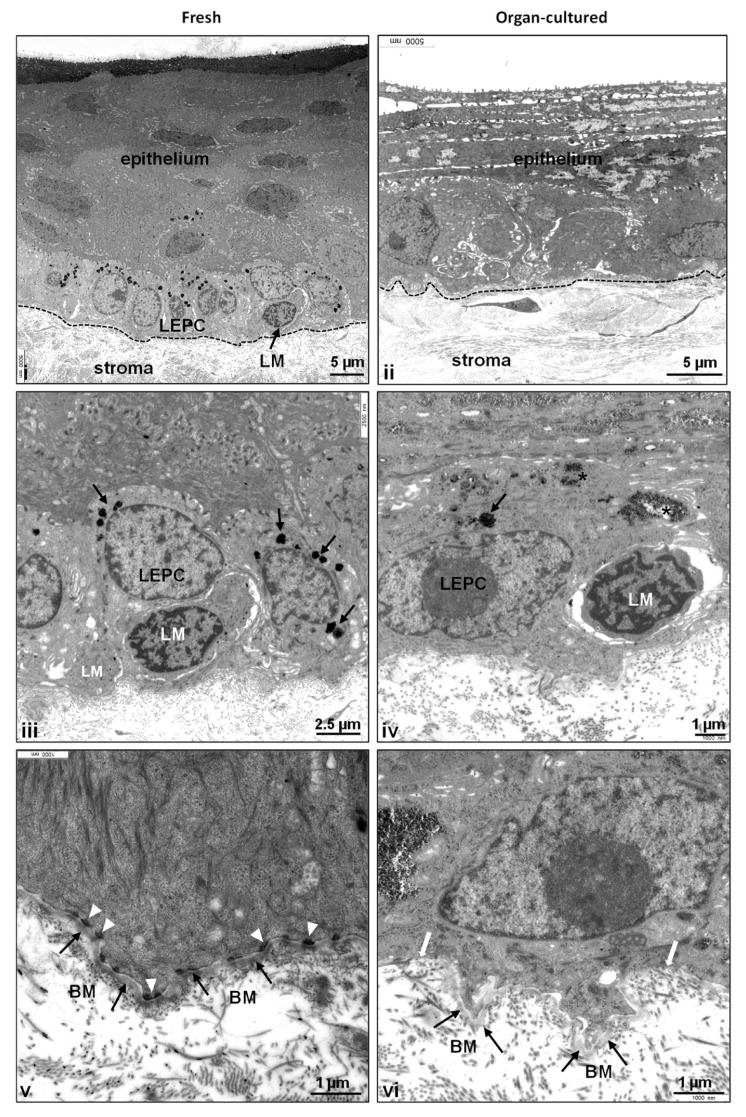
Ultrastructural analysis of organ-cultured human corneal limbus: The transmission electron microscopy analysis of limbal tissues showing a well-defined 8–10 layered limbal epithelium, characterized by clusters of small, roundish densely packed cells situated at the base of the epithelial papillae of fresh tissue (**i**); the organ-cultured limbal tissue consists of a thinned limbal epithelium (3–4 layers) with basal cuboidal cells along with flattened superficial cells and no signs of palisade structures ((**ii**), dashed line represents the basement membrane (BM)); a frequent accumulation of glycogen (asterisk, (**iv**)) and remnants of melanosomes (arrows) within the basal epithelial cells of organ-cultured tissue (**iv**) in contrast to fresh tissues (**iii**); basal adhesion of LEPC mediated by hemidesmosomes (white arrowheads) to the BM (black arrows) in fresh tissue (**v**); gaps in the BM of organ-cultured limbal tissue (arrows, (**vi**)). Abbreviations: LEPC, limbal epithelial progenitor cells; LM, limbal melanocytes.

**Figure 3 ijms-24-16856-f003:**
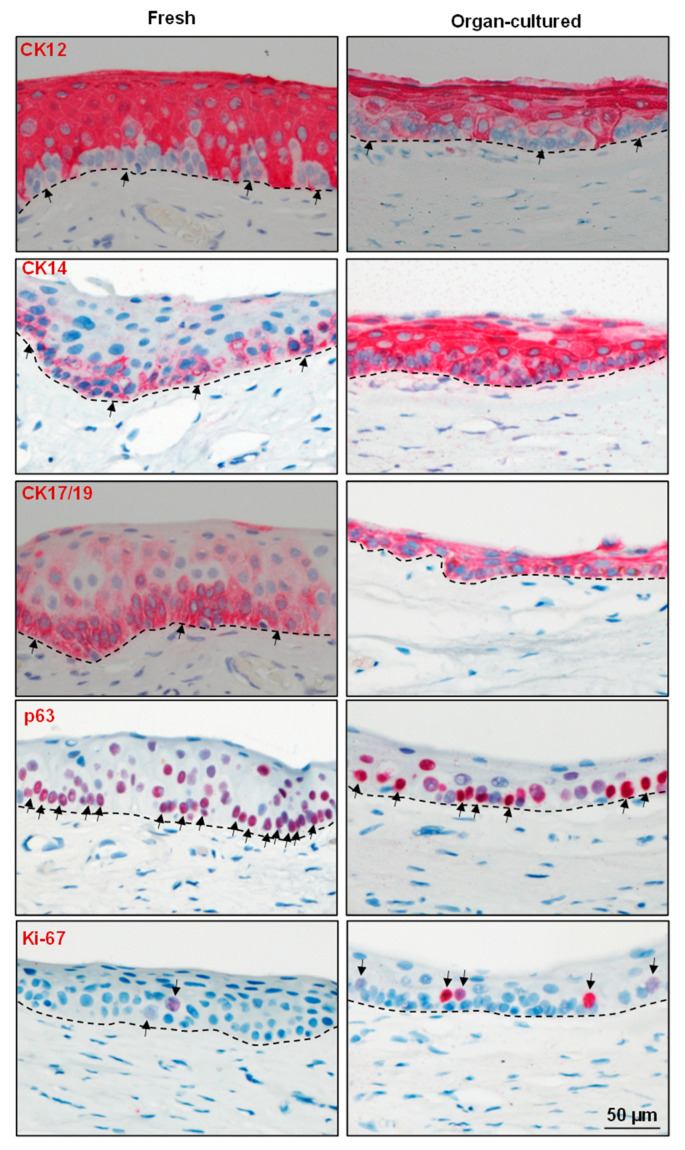
Immunohistochemical analysis of organ-cultured human corneal limbus: Immunohistochemical analysis of limbal tissue sections showing the expression of cytokeratin (CK) 12 (red) in suprabasal cells of both organ-cultured and fresh tissues but not in the basal cell layers (arrows); CK14 and CK17/19 (red) expression in all layers of limbal epithelium in the organ-cultured limbal tissue, whereas it is limited to basal cells in the fresh limbal tissue (arrows); p63α and Ki-67 positive cells (arrows, red) in the basal layers of limbal epithelia, but the density of positive cells are low in organ-cultured limbal tissues. Dased line represents the basement membrane.

**Figure 4 ijms-24-16856-f004:**
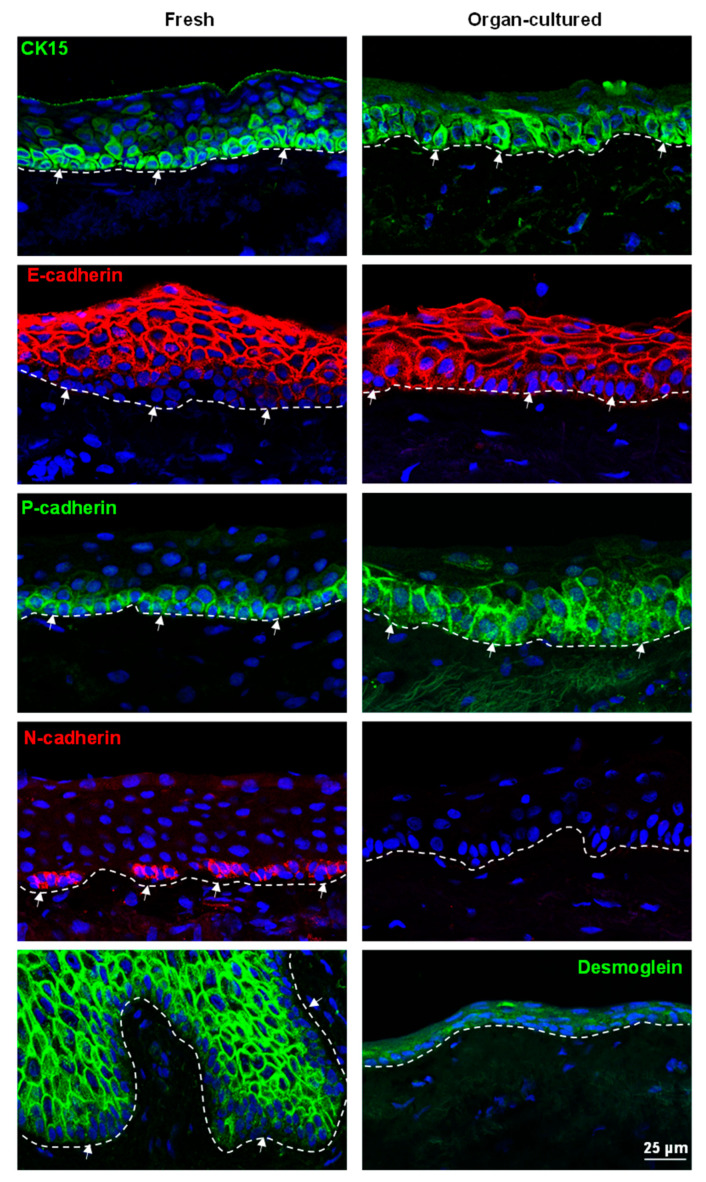
Immunohistochemical analysis of organ-cultured human corneal limbus: Immunohistochemical analysis of limbal tissues showing the CK15 (green, arrows) expression in the basal layers of the limbal epithelia in both organ-cultured and fresh tissues but rather weak in the organ-cultured limbal tissues; E-cadherin (red) expression in the superficial layers of limbal epithelial cells in both the organ-cultured and fresh limbal tissues with rather weak/no expression in the basal cells (arrows); P-cadherin (green, arrows) expression in basal layers of limbal epithelium in both fresh and organ-cultured tissues; N-cadherin (red, arrows) expression in the basal cells of fresh limbal tissue, but absent in organ-cultured limbal tissues; desmoglein expression (green) in the suprabasal layers of epithelia but not in the basal layers (arrows) in fresh tissues, whereas its expression is absent in the organ-cultured limbal tissues. Dashed line represents the basement membrane. Nuclear counterstaining with 4′,6-diamidino-2-phenylindole (blue).

**Figure 5 ijms-24-16856-f005:**
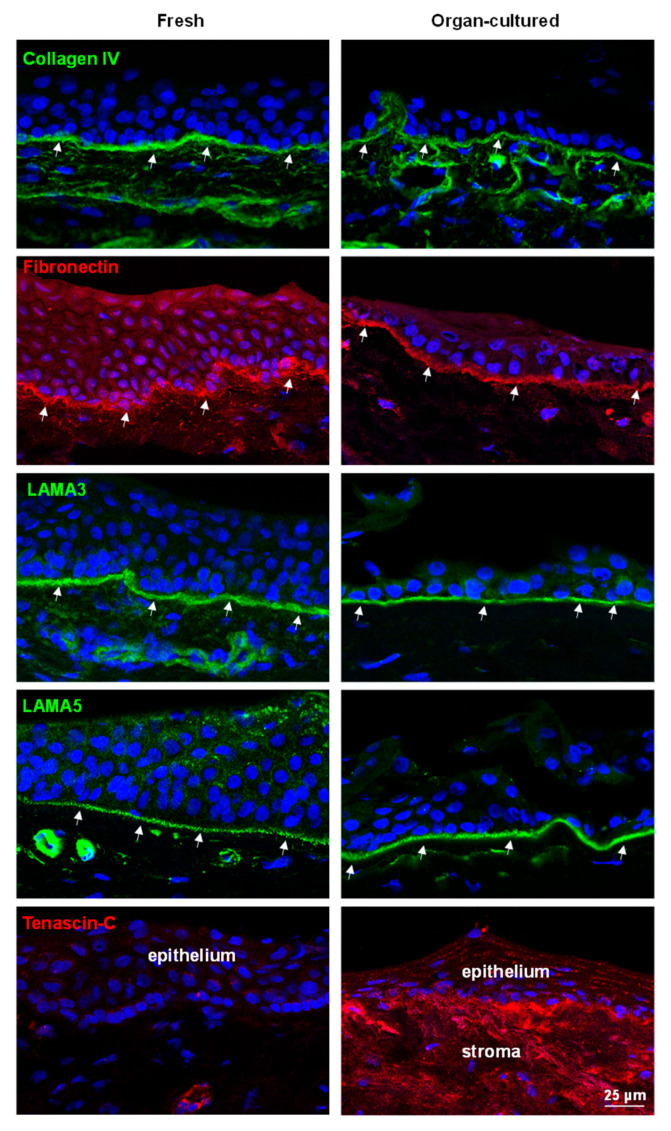
Immunohistochemical analysis of organ-cultured human corneal limbus: Immunohistochemical analysis of limbal tissue sections showing the collagen IV (green, arrows) expression in the basement membrane (BM) of the limbal epithelium as well as in vessels of both tissues; fibronectin (red, arrows) expression in the limbal BM of both tissues; laminin alpha 3 and alpha 5 expression (green, arrows) in the BM of both limbal tissues; Tenascin-C expression (red) beneath the limbal epithelium, as well as in the limbal stroma of organ-cultured tissues but not in the fresh tissues. Nuclear counterstaining with 4′,6-diamidino-2-phenylindole (blue).

**Figure 6 ijms-24-16856-f006:**
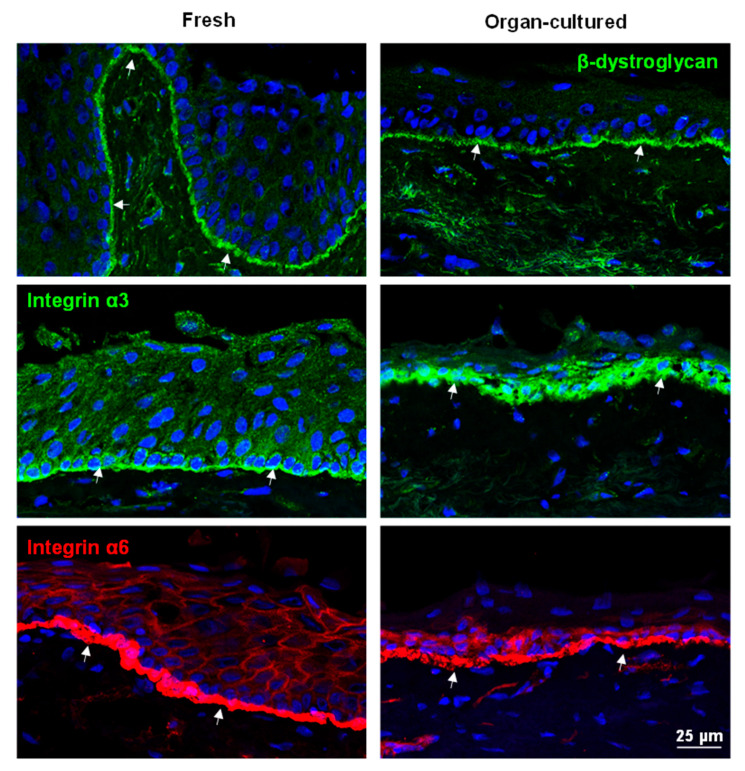
Immunohistochemical analysis of organ-cultured human corneal limbus: Immunohistochemical analysis of limbal tissue sections showing the β-dystroglycan (green, arrows) expression in the basal cell membranes of the limbus as well as in the stroma of both organ-cultured and fresh tissues; integrin α3 staining (green, arrows) in the basal aspect of the limbal epithelia of fresh limbal tissue, whereas its expression present in the basal and lateral aspects of limbal epithelia of organ-cultured limbal tissue; integrin α6 (red, arrows) expression in all limbal basal epithelia of fresh limbal tissue with a continuous strong basal membrane staining, rather weak in the organ-cultured limbal tissues. The dashed line represents the basement membrane. Nuclear counterstaining with 4′,6-diamidino-2-phenylindole (blue).

**Figure 7 ijms-24-16856-f007:**
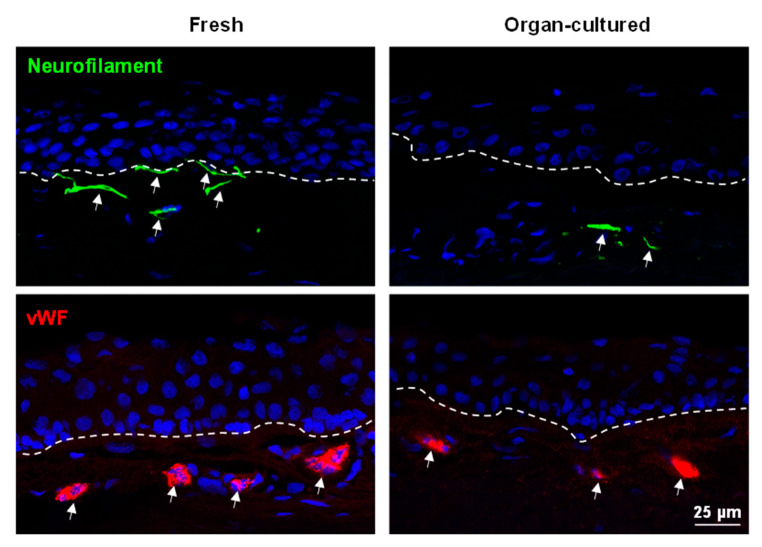
Immunohistochemical analysis of organ-cultured human corneal limbus: Immunohistochemical analysis of limbal tissue sections neurofilament-L expression (arrows, green) in the stroma of fresh limbal tissues, but rarely present in organ-cultured limbal tissues; von Willebrand factor (vWF) expression in the vessels of both fresh and organ-cultured limbal tissues. The dashed line represents the basement membrane. Nuclear counterstaining with 4′,6-diamidino-2-phenylindole (blue).

**Figure 8 ijms-24-16856-f008:**
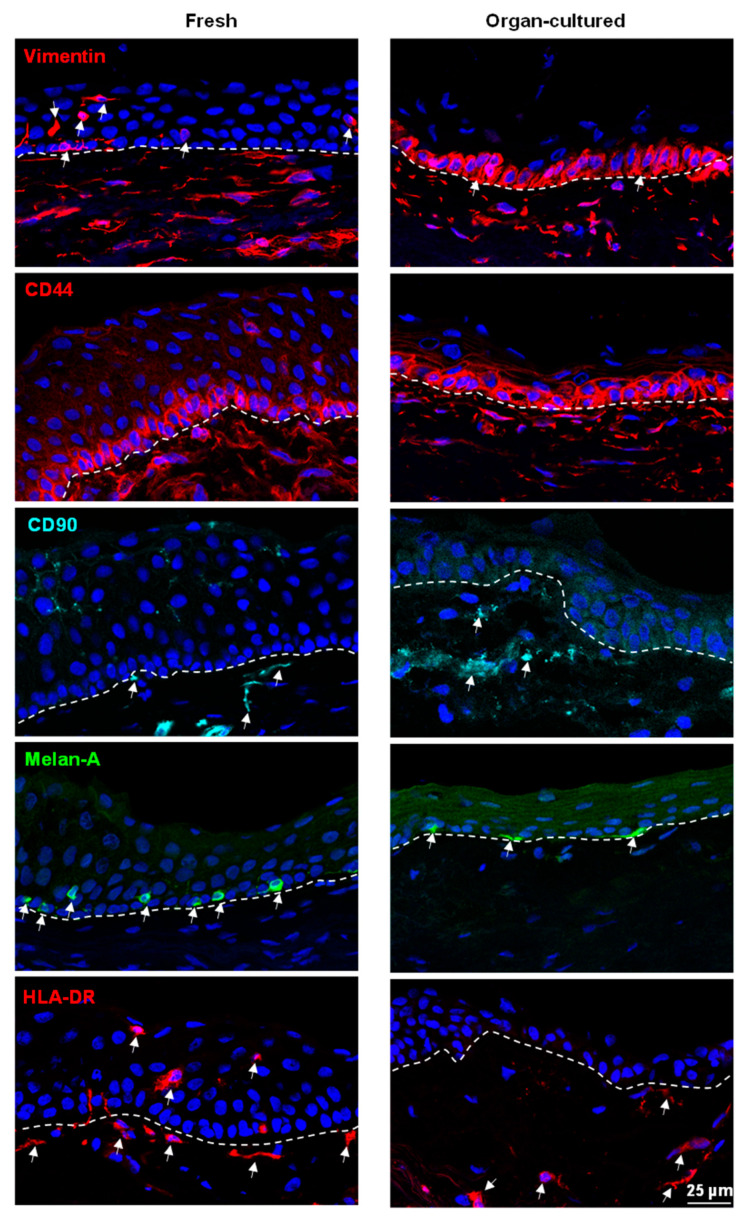
Immunohistochemical analysis of organ-cultured human corneal limbus: Immunohistochemical analysis of limbal tissue showing the vimentin (red) expression in the limbal stroma of both organ-cultured and limbal tissues as well as in the basal limbal epithelium of organ-cultured tissues (arrows), vimentin^+^ cells in the limbal epithelium of fresh tissues (arrows); CD44 (red) expression in the basal cells of limbal epithelia, stroma, and vasculature of both limbal tissues; CD90^+^ cells (arrows, cyan) in the stroma of both fresh and organ-cultured limbal tissues; melan-A positive melanocytes (green) in the basal layers of limbal epithelium of both limbal tissues; HLA-DR^+^ cells in both the limbal epithelium (red, arrows) and stroma of fresh tissues (arrows), but only present in the limbal stroma of organ-cultured tissue. Dashed line represents the basement membrane. Nuclear counterstaining with 4′,6-diamidino-2-phenylindole (blue).

## Data Availability

The datasets generated during and/or analyzed during the current study are available from the corresponding author upon reasonable request.

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
