# Peer review of "Influence of Organ Culture on the Characteristics of the Human Limbal Stem Cell Niche"

_ijms, 2023, doi:10.3390/ijms242316856_

Round 1

Reviewer 1 Report

Comments and Suggestions for Authors

The authors of the article entitled "Influence of organ culture on the characteristics of the human limbal stem cell niche" have carried out a high quality work comparing the expression of several limbal markers between fresh sclerocorneal limbus and those maintained under transport conditions for a certain period of time. The data yield interesting results in terms of altered expression of some key molecules crucial for limbal maintenance, adding to existing knowledge. Perhaps, the quality of the work could be enhanced if the authors compared the expression of these markers between different methods of limbal storage. 

However, some minor suggestions could be:

-The authors could consider evaluating the ABCG2 marker, another commonly associated marker with limbal epithelial progenitor cells (LEPC).

-The distribution of the images related to neurofilament and von Willebrand factor does not align with their description in the text. They appear in a subsequent and separate section from the rest of the content in Figure 6 (distroglycan and integrins). Therefore, it might be more appropriate to organize them into a panel for basement membrane components (Col IV, fibronectin, laminins, dystroglycan, and integrins) and another for stromal components (tenascin, neurofilament, and von Willebrand). Alternatively, considering placing neurofilament and von Willebrand in a separate or supplementary image.

Reviewer 2 Report

Comments and Suggestions for Authors

The current manuscript aims to examine the effect of organ culture on the characteristics of the human limbal stem cell niche. Although the topic is interesting in its scientific field, there are some issues that require the authors’ attention to improve the quality of this particular manuscript before further consideration for publication in a high-quality journal “IJMS”.

Specific comments:

1.         The authors should carefully clarify the differences in the academic contribution points between the current manuscript and their earlier reports (please refer to the following papers: #1 DOI: 10.1167/iovs.16-19354 & #2 10.3390/ijms24087543). In particular, the characteristics of the human limbal stem cells obtained from stored organ-cultured tissue have previously been explored. Please justify.

2.         Figure 1A identified the vasculature features in the histology, but relevant discussion was absent. Please improve.

3.         As stated by the authors, various studies have reported that the LEPC express CK15, CK14, CK17/19, p63, P(placental)-cadherin, N(neural)-cadherin, while being negative for CK12, CK3. However, the data presentation about CK3 expression level is missing in Figure 3 and Figure 4. Please improve.

4.         Furthermore, Figure 3 presented the data of Ki-67 staining, but relevant results and discussion were absent. Please improve.

5.         According to my observations on the data presentation in Figure 4, no obvious difference in the epithelial thickness could be found between the fresh tissue and organ-cultured counterparts. However, the histological imaging data presented in other figures did not support the aforementioned finding. Why? Please justify.

6.         As stated by the authors, ECM molecules have a significant role as they instruct stem/progenitor cell signaling and homeostasis. However, such an important claim was not supported by any documented reference. In order to balance scientific viewpoint, the authors are highly recommended to consider the inclusion of this supportive case study (DOI: 10.1016/j.actbio.2015.11.042) in the reference list.

Reviewer 3 Report

Comments and Suggestions for Authors

Organ culture storage techniques for corneoscleral limbal (CSL) tissue have improved the quality of corneas for transplantation and allow for longer storage times. Cultured limbaltis sue has been used for stem cell transplantation to treat limbal stem cell deficiency (LSCD) as well as for research purposes to assess homeostasis mechanisms in the limbal stem cell niche. However, the effects of organ culture storage conditions on the quality of limbal niche components are less well described. The rusults in this study shows that the organ culture-induced alterations of the limbal epithelial stem cell niche might hamper its use in the treatment of LSCD as well as in research studies. In contrast, reduced numbers of donor-derived Langerhans cells seem associated with better clinical outcomes. However, there is a need to consider the preferential use of fresh CSL for limbal transplants and to look at ways of improving the limbal stem cell properties of stored CSL tissue. The findings in this study is of great use for the evaluate and application of Organ culture storage techniques for corneoscleral limbal (CSL) tissue, as well as the mechanism that immune rejection of tranlplanted cornea limbus prevension. The manuscript is well written, figures are clear. 

Round 2

Reviewer 2 Report

Comments and Suggestions for Authors

The authors’ revision is highly appreciated. However, after my careful reading and checking, one minor issue still requires the authors’ attention to improve the quality of this particular manuscript for publication in a high-quality journal ”IJMS”.

Original Comment #6

As stated by the authors, ECM molecules have a significant role as they instruct stem/progenitor cell signaling and homeostasis. However, such an important claim was not supported by any documented reference. In order to balance scientific viewpoint, the authors are highly recommended to consider the inclusion of this supportive case study (DOI: 10.1016/j.actbio.2015.11.042) in the reference list.

Authors’ Response #6

We agree with the reviewer and the suggested the reference has been included in the study.

Unsolved Comment #6

According to my checking, this specific citation is missing in the revised manuscript. If possible, please consider the suggestion to include this relevant article in the authors’ contribution to update the manuscript content.

Author Response

Thank you. The mentioned reference is already included in the text as reference number 51(line 281) and also in the reference list as

Preservation of Human Limbal Epithelial Progenitor Cells on Carbodiimide Cross-Linked Amniotic Membrane via Integrin-Linked Kinase-Mediated Wnt Activation - ScienceDirect Available online: https://www.sciencedirect.com/science/article/pii/S174270611530218X?via%3Dihub (accessed on 22 November 2023).